# Relationship between Metabolomics Profile of Perilymph in Cochlear-Implanted Patients and Duration of Hearing Loss

**DOI:** 10.3390/metabo9110262

**Published:** 2019-11-01

**Authors:** Thuy-Trân Trinh, Hélène Blasco, Patrick Emond, Christian Andres, Antoine Lefevre, Emmanuel Lescanne, David Bakhos

**Affiliations:** 1Service ORL et Chirurgie Cervico-Faciale, CHRU de Tours, 37000 Tours, France; lescanne@univ-tours.fr (E.L.); david.bakhos@univ-tours.fr (D.B.); 2Laboratoire de Biochimie et Biologie Moléculaire, CHRU de Tours, 37000 Tours, France; helene.blasco@univ-tours.fr (H.B.); andres@med.univ-tours.fr (C.A.); 3Université François-Rabelais, 37000 Tours, France; patrick.emond@univ-tours.fr; 4Inserm U1253, 37000 Tours, France; antoine.lefevre@univ-tours.fr; 5Centre SLA, Service de Neurologie, CHRU Bretonneau, 37044 Tours, France; 6PPF (Programme Pluri-Formation), Université François-Rabelais, 37000 Tours, France

**Keywords:** cochlear implant, metabolomics, perilymph, duration of hearing loss

## Abstract

Perilymph metabolomic analysis is an emerging innovative strategy to improve our knowledge of physiopathology in sensorineural hearing loss. This study aims to develop a metabolomic profile of human perilymph with which to evaluate the relationship between metabolome and the duration of hearing loss. Inclusion criteria were eligibility for cochlear implantation and easy access to the round window during surgery; patients with residual acoustic hearing in the ear to be implanted were excluded. Human perilymph was sampled from 19 subjects during cochlear implantation surgery. The perilymph analysis was performed by Liquid Chromatography−High-Resolution Mass and data were analyzed by supervised multivariate analysis based on Partial Least-Squares Discriminant Analysis and univariate analysis. Samples were grouped according to their median duration of hearing loss. We included the age of patients as a covariate in our models. Statistical analysis and pathways evaluation were performed using Metaboanalyst. Nineteen samples of human perilymph were analyzed, and a total of 106 different metabolites were identified. Metabolomic profiles were significantly different for subjects with ≤12 or >12 years of hearing loss, highlighting the following discriminant compounds: *N*-acetylneuraminate, glutaric acid, cystine, 2-methylpropanoate, butanoate and xanthine. As expected, the age of patients was also one of the main discriminant parameters. Metabolic signatures were observed for duration of hearing loss. These findings are promising steps towards illuminating the pathophysiological pathways associated with etiologies of sensorineural hearing loss, and hold open the possibilities of further explorations into the mechanisms of sensorineural hearing loss using metabolomic analysis.

## 1. Introduction

Sensorineural hearing loss (SNHL) can negatively affect the development of spoken language, education and social interactions, can reduce quality of life, and limit human communication and socio-professional relations [1]. Around 5% of the worldwide population suffers from different degrees of hearing loss [2], and SNHL is the most common sensory deficit in more developed countries [3,4]. Causes of SNHL have been described, and can be congenital [5,6] or acquired [7]. In the cases of patients with severe-to-profound SNHL who receive no benefit from hearing aids, cochlear implants (CIs) can partially restore hearing by electrically stimulating the surviving auditory nerve fibers [8].

Two of the strongest predictors of CI outcomes are the etiology and duration of deafness [9,10]. Although the etiology of SNHL may be evident for some patients, progressive hearing loss remains an enigma. SNHL may be caused by damage to approximately 30 different cell types, and by alteration of the cochlear nerves. Often, the site of this cochlear damage is unclear, and cannot be determined from clinical or radiological measures. From a histological point of view, the most common mechanism is hair cell (HC) loss and damage to the spiral ganglion neurons (SGNs).

Cellular mechanisms involved in SNHL are intricately linked. Five main pathways are described by Wong et al. [11]: (1) Accumulation of reactive oxygen species (ROS) and reactive nitrogen species (RNS) in the cochlea is a source of oxidative stress that contributes to apoptosis through a cascade of reactions with DNA, proteins and cytosolic molecules. (2) An increase of free Ca^2+^ in the HCs is also a source of mitochondrial disturbance, apoptosis and HC death. (3) MAPKs are signaling proteins contributing to the stress-activated protein kinase, a key mediator in oxidative stress and inflammation. (4) Programmed HC death can be mediated by the evoked intracellular pathway or by the extracellular pathway caspase. (5) Infection and inflammation may also induce apoptosis through a response based on innate immunity, promoting the death of pathogens. While many pathways leading to HC and SGN loss have been described and appear to be interlinked, no diagnostic tool is currently available to characterize the lesion site within the cochlea and intracellular mechanisms of SNHL.

Auditory and language performance with CIs also depend upon the duration of deafness [9]. In cases of late cochlear implantation, performance is generally poor due to delayed auditory cortical maturation and cortical reorganization [12,13]. In this context, a reliable tool with which to assess the etiology and the duration of deafness would help to improve deafness management. Better understanding of the pathophysiological mechanisms of SNHL would allow for better characterization of the role of cochlear cells in the progression of hearing loss.

Metabolomics is an emerging strategy that can be used to find biomarkers that are useful in increasing the knowledge of disease pathogenesis, and may help in diagnosis or prognosis. The development of high throughput analytical methods such as Liquid Chromatography, coupled with High Resolution Mass Spectrometry (LC-HRMS) has enabled numerous health applications for an analysis of biological fluids, cells and tissue [14]. In the inner ear, HCs are fully immersed in the perilymph, as are the cell bodies of SGNs. Thus, it is possible that the perilymph contains molecules released by cells involved in SNHL [15,16]. The exploration of this fluid may help to better understand cells’ death from sensory organs, and may be useful to characterize different types and durations of hearing loss. This technical analysis on a low volume of perilymph sample has already been validated in a methodological study [17]. Metabolomics analysis on perilymph have been reported on guinea pigs to compare perilymph metabolome before and after induction of deafness. Fujita et al. [18] highlighted 10 metabolites with a different concentration before and after noise exposure, while Fransson et al. [19] did a more descriptive analysis, and showed that the multivariate analysis showed a good model to separate the two groups (ototoxicity induced by cisplatin and cisplatin+H2), thanks to the metabolomic profile of perilymph.

The objective of this study was to describe the metabolomic profile of perilymph extracted from patients with SNHL during CI surgery.

## 2. Results

### 2.1. Metabolites Analysis

In total, 106 different metabolites were identified (Appendix A). No outliers were identified on the PCA score plot for quality control samples and patients.

### 2.2. Significant Metabolic Profile for Duration of Hearing Loss

Patients were divided into two groups according to the median duration of hearing loss (12 years). Group characteristics are presented in Table 1. The mean duration of hearing loss was 5 ± 4 years for Group 1 (*n* = 10) and 36 ± 15 years for Group 2 (*n* = 9). The mean age at cochlear implantation was 38 ± 30 years for Group 1 and 65 ± 19 years for Group 2; a Mann-Whitney test showed no significant difference between the groups for age at implantation (*p* = 0.062). As different etiologies were observed in the cohort, PCA was used to evaluate the distribution of etiologies within each group; and within each group, the etiologies were similarly distributed.

Supervised multivariate analysis revealed a good model to discriminate Group 1 and 2 according to the following metabolites (accuracy 63%, Figure 1): *N*-acetylneuraminate, glutaric acid, l-cystine, 2-methylpropanoate, butanoate, xanthine, l-histidine, S-lactate, 4-hydroxy-l-proline, serotonin, 2-deoxy-D-glucose, *N*-acetyl-l-alanine, l-proline and taurine. As expected, age of patients was the second most important discriminant parameter in the model.

The volcano plot highlighted only one metabolite (*N*-acetylneuraminate) with *p* < 0.1 and fold change (Group 1, Group 2) > 1.2. All metabolites included in multivariate model were also relevant in univariate model but did not reach significance.

## 3. Discussion

In this study, metabolomic analyses were performed on perilymph samples collected from patients with SNHL during a cochlear implantation to describe a metabolomic profile according to duration of hearing loss.

### 3.1. Innovative Exploration of Perilymph Content

We demonstrated that such an approach is possible, despite difficulties in obtaining a sufficient volume of human perilymph. For two of the recruited but excluded patients, it was not possible to collect a perilymph sample due to poor exposure of the round window. As described in Schmitt et al. [16], a highly cautions approach must be used for an accurate sampling of perilymph during inner ear surgery to avoid intraoperative contaminations. To sample the perilymph, we used a 22 g needle. Other researchers have used a modified micro glass capillary in order to minimize inner ear trauma during perilymph collection, especially for patients with residual hearing [16]. In this study, such patients with residual hearing were excluded.

Perilymph analysis remains a major challenge. Since 1966, researchers have been analyzing the perilymph profile to increase knowledge of hearing loss, beginning with animals. Since then, proteomic studies on human perilymph have been performed successfully [15,16,20,21,22]. Thalmann et al. [21] were the pioneers in this field, using high-resolution, two-dimensional electrophoresis to study protein profiles in human perilymph. Fujita et al. [18] performed metabolomic analysis of perilymph collected from guinea pigs using gas chromatography coupled to mass spectrometry; they detected 77 different metabolites, and found that while the metabolomic profile for perilymph changed before and after noise exposure, the metabolomic profile for blood plasma did not change. The concentration of ten metabolites were significantly different before and after noise exposure: 3-hydroxy-butyrate, glycerol, fumaric acid, galactosamine, pyruvate + oxalacetic acid, phosphate, meso-erythritol, citric acid + isocitric acid, mannose and inositol. More recently, Schmitt et al. [22] sampled human perilymph during inner ear surgeries and analyzed the proteomic profile. They identified heat shock proteins (HSPs), known for their anti-apoptotic and anti-necrotic effects.

To our knowledge, this work is the first to use metabolomic analysis performed on human perilymph samples to correlate with clinical criteria. This methodological analysis has already been validated in Mavel et al. 2018 [17]. This approach may shed new light on pathophysiological pathways, as it illuminates patterns of small molecules that may relate to energetic metabolism disturbance, or could be precursors of actors involved in inflammation, oxidative stress, etc. We used the most sensitive technique to analyze a small volume of perilymph (LC-HRMS), and were able to identify 106 metabolites that covered more than 10 metabolic pathways. We analyzed the relationships between metabolites and metabolic pathways to their duration of hearing loss. We were able to identify metabolites and associated pathways altered in some phenotypes of patients according to this duration of hearing loss.

### 3.2. Perilymph Metabolome Discriminates Patients According to Hearing Loss Duration

The discrimination using the duration of hearing loss as a parameter permitted to well discriminate two groups. However, we noticed some variability for the patients included in each group. This result can underlie an interindividual variability. We may suspect that this observation is due to the high heterogeneity of patients within each group that is mainly explained by the different etiologies. Etiology was not thoroughly investigated in this study, given the distribution of etiologies within the relatively small number of subjects. The size of the cohort may also suggest a lack of robustness of the model of prediction.

Whatever, the impact of metabolites in the model remains relevant, and has to be specified in an analysis on a larger cohort with the inclusion of all sources of heterogeneity in the models.

Unsurprisingly, the age of patients (i.e., the effective age of the perilymph sample) plays a key role in the discrimination between groups according to hearing loss duration.

### 3.3. Metabolites and Associated Metabolic Pathways Involved in Hearing Loss Duration

We found a correlation between *N*-acetylneuraminate and the duration of hearing loss. This metabolite is the predominant sialic acid found in mammalian cells, is generally located at the terminal of glycoprotein and glycolipid on the surface of the cell membrane, and is a product of glucose degradation. Elevated rates of this metabolite may be explained by the cell membrane breaking (i.e., HC apoptosis). We would expect elevated levels of this metabolite during ongoing cell death, as in the beginning of progressive hearing loss, and not after a long duration of hearing loss. This hypothesis is difficult to validate, as the direct account of survival hair cells was not performed in this study. Histological study will be necessary to confirm this hypothesis.

### 3.4. Perspectives of Perilymph Metabolome

By revealing molecular mechanisms of hearing loss and understanding the role of each metabolite in the intricate auditory network, deafness endophenotypes may be better characterized according to metabolic signature, allowing for better understanding of the different pathways involved in hearing loss. The approach is promising, but needs further studies with more patients, especially to investigate the relationship between perilymph metabolomics profiles and SNHL etiologies. A multi-center study would be preferable to recruit adequate numbers of patients with a distribution of etiologies, and will ultimately provide a better understanding of the physiopathology of SNHL. Animal studies, where the age at hearing loss, severity of hearing loss, age at implantation and chronological age can be explicitly controlled, may also provide important insights regarding how these factors may be reflected in the metabolomic profile.

## 4. Materials and Methods

### 4.1. Patients

Subjects were recruited from the otolaryngology department at University Hospital of Tours. All participants provided written informed consent. The Ethics Committee of the University Hospital of Tours approved the protocol (2015-045). All the patients met criteria for a cochlear implantation. Any patient with a residual acoustic hearing in either ear was excluded from the study. Twenty-one patients who underwent CI surgery from June 2015 to July 2016 were included in the study. Two patients were subsequently excluded during CI surgery due to a poor exposure of the round window (*n* = 2). Thus, 19 patients were ultimately included in the study.

Demographic information such as sex, age, onset of hearing loss, duration of hearing loss, duration of hearing aid (HA) use and etiology of hearing loss, was collected as part of the standard CI candidacy evaluation (Table 2). Among the 19 CI patients who were included in the analysis, 8 were women and 11 were men. The age at testing ranged from 4 to 86 years (mean = 51 ± 28), onset of hearing loss ranged from 0 (congenital) to 67 years (median = 29; mean = 31 ± 28), duration of hearing loss ranged from 0.5 to 67 years (median = 12; mean = 20 ± 19) and duration of HA use before cochlear implantation ranged from 0 to 45 years (median = 7; mean = 16 ± 16). Six etiologies of hearing loss were identified: Congenital (*n* = 8), presbycusis (*n* = 5), sudden hearing loss (*n* = 2), temporal bone fracture (*n* = 2), Meniere’s disease (*n* = 1), trauma (*n* = 1). Audiometric data was collected for at least at one year before implantation, shortly before implantation, and one year after implantation. Before implantation, aided word recognition was <50% correct at 60 dB SPL for all the included patients.

Patients were grouped according to the median duration of hearing loss (≤12 y; >12 y) (Table 1). This parameter was selected instead of etiology due to the large heterogeneity of etiology among patients. Note that the duration of hearing loss may represent some period of moderate-to-severe hearing loss before subjects were considered CI candidates (i.e., severe-to-profound deafness). As shown in Table 2, some subjects used hearing aids (HAs) for some period prior to cochlear implantation.

### 4.2. Sample Collection

Perilymph samples were collected from patients during CI surgery, before parenteral injection of solumedrol 1mg/kg. After the posterior tympanotomy, the round window was exposed. We collected around 1 µL of perilymph using a 22 g needle through the round window membrane. The perilymph samples were stored in polypropylene tubes at −80 °C immediately after collection and until analysis. Before testing, samples were thawed, centrifuged at 3000× *g* for 5 min and 0.8 µL were aliquoted.

The metabolites found in the perilymph and the metabolic pathways were analyzed for all patients.

### 4.3. Liquid Chromatography−High-Resolution Mass Spectrometry

LC-HRMS analysis was performed using a UPLC Ultimate 3000 system (Dionex, Mainz, Germany), coupled to a Q-Exactive mass spectrometer (Thermo Fisher Scientific, Bremen, Germany) and operated in the positive (ESI+) and negative (ESI−) electrospray ionization modes (one run for each mode). The perilymph analysis method, including the protocols for metabolite extraction and MS methods, are fully described in Mavel et al. [17]. Liquid chromatography was performed using a Phenomenex Kinetex 1.7 μm XB – C18 column (100 mm × 2.10 mm) maintained at 40 °C and a hydrophilic interaction liquid chromatography (HILIC) column (100 mm × 2.10 mm, 100 Å). The injection volume used was 10 µL.

Metabolism of perilymph was quenched and metabolites were extracted with 100 µL of methanol added to 0.8 µL of perilymph. Samples were vortexed for 1 min and stirred for 10 min at 4 °C. The samples were left at −20 °C for 30 min then centrifuged at 15,000× *g* for 15 min at 4 °C. The supernatant (95 µL) was evaporated with a SpeedVac concentrator at 40 °C and the dry residue was re-suspended in 100 µL of methanol/water (50:50). Finally the samples were vortexed for 1 min and centrifugated at 20,000× *g* at 4 °C for 10 min. Quality control (QC) samples were prepared from a mix of all extracted samples.

Metabolites were identified with a library of standard compounds (Mass Spectrometry Metabolite Library^®^, St. Louis, MO, USA) and according the retention time (± 20 s of the standard reference), the molecular mass (within a range of 10 ppm around the reference) and the isotopic ratios. The signal value was calculated using Xcalibur^®^ software version 4.1 (Thermo Fisher 13 Scientific, San Jose, CA, USA) by integrating the chromatographic peak area corresponding to the selected metabolite.

Metabolites levels were normalized by the sum of all the metabolites of the profile, then data were log-transformed and autoscaled (mean-centered and divided by the standard deviation of each variable).

### 4.4. Univariate Analysis

The univariate analysis was based on the non-parametric Wilcoxon test using Metaboanalyst (V 2.1). Univariate analysis is represented by a volcano plot combining two relevant information: Fold-change and *p*-value of the *t*-tests. The x-axis represents the fold change between the subject groups (log scale). The y-axis represents the *p*-value for *t*-tests of differences between samples (negative log scale). Correction for multiple test was systematically applied, but results without correction were presented to have an overview of the most promising metabolites in order to give more opportunity to explain pathophysiology with no aim of prediction.

### 4.5. Multivariate Analysis

The classification method was based on unsupervised Principal Component Analysis (PCA) to evaluate distribution of samples and identify outsiders, and supervised analysis based on Partial Least-Squares Discriminant Analysis (PLS-DA) with associated accuracy prediction. Values of Variable Influence on Projection (VIP) represent the importance of the compound (metabolite/lipid) for the PLS-DA models, the score plots show the classified samples, and the loadings characterize the relation between the Y and X variables (metabolites, lipids). We included age of patients as a covariate in our models.

## 5. Conclusions

This work provides proof of concept for a promising new strategy with which to better understand hearing loss. We used novel approaches toward collecting and analyzing perilymph in patients with SNHL, and found a correlation between the metabolomic profiles of perilymph and the duration of hearing loss.

## Figures and Tables

**Figure 1 metabolites-09-00262-f001:**
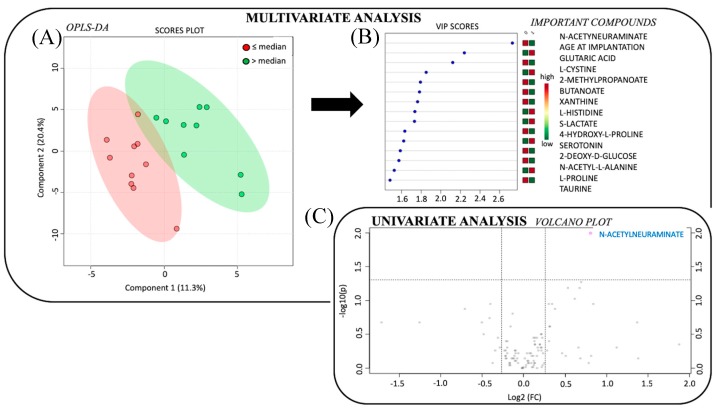
Statistical analysis to compare the metabolome profile from perilymph fluid of cochlear implants (CIs) patients according to duration of hearing loss. (**A**) Multivariate analysis by partial least squares discriminant analysis (PLS-DA), discriminating between patients according to the median duration of hearing loss. The red circles show data for patients with duration of hearing loss ≤ 12 years, and the green circles show data for patients with duration of hearing loss > 12 years. Components 1 and 2 represent a linear combination of relevant metabolites expressing the maximum variance. After mean-centering and scaling to unit variance, the data are used for the computation of the first principal component, that is the line in the K-dimensional space that best approximates the data in the least squares sense. Importantly one principal component is insufficient to model the systematic variation of a data set, and a second principal component is calculated. The second PC is also represented by a line in the K-dimensional variable space, which is orthogonal to the first PC. (**B**) The rank of the different metabolites (the top 15) identified by the PLS-DA according to the VIP score on the left and according to the coefficient score on the right. (**C**) Univariate analysis via volcano plot based on fold change and p-value, highlighting 1 metabolite. Legend: VIP: Variable Influence of Projection; OPLS-DA: Orthogonal Partial Least Squares Discriminant Analysis; p: *p*-value; FC: fold change.

**Table 1 metabolites-09-00262-t001:** Characteristics of the patients grouped according to the median duration of deafness (12 years).

	Group 1 (≤12 y)	Group 2 (>12 y)	p; U
N	10	9	
Mean age at CI (y)	38 ± 30	65 ± 19	*p* = 0.062; U = 22.0
Mean age onset of deafness (y)	33 ± 30	28 ± 26	*p* = 0.733; U = 40.5
Mean duration of deafness (y)	5 ± 4	36 ± 15	*p* < 0.001; U = 0.0 *
Mean duration of HA use (y)	3 ± 2	20 ± 13	*p* < 0.001; U = 0.0 *
Pre-surgical PTA (dB)	98 ± 16	91 ± 14	*p* = 0.561; U = 37.5
SAT (dB)	>100 dB	>100 dB	

CI: Cochlear implantation; y = years; HA: Hearing aids; PTA = pure tone average threshold across 500, 1000, 2000 and 4000 Hz; SAT: Speech audibility threshold (dB threshold for 50% word-in-sentence recognition). Asterisks indicate significant differences between groups.

**Table 2 metabolites-09-00262-t002:** Demographic information for CI patients.

Subjects	Sex	Age at CI (y)	Age at Onset of Deafness (y)	Duration of Deafness (y)	Duration of HA (y)	Pre-Surgical PTA (dB)	Etiology	Site of CI	Type of CI	Comorbidities
1	F	39	2	37	37	81.25	Congenital	Left	Cochlear CI 512	#
2	M	86	26	60	22	81.25	Traumatic	Left	Cochlear CI 512	#
3	M	63	53	10	1	85	Meniere’s	Right	Cochlear CI 512	#
4	F	9	2	7	7	78.5	Congenital	Right	MedEl Synchrony Pin	#
5	F	79	67	12	4	116.25	Temporal bone fracture	Left	Cochlear CI 512	High blood pressure
6	M	63	62	0.5	5	116.25	Sudden HL	Left	Cochlear CI 512	#
7	F	42	2	40	40	83.75	Congenital	Right	Cochlear CI 512	#
8	F	59	58	1	1	110	Sudden HL	Right	Cochlear CI 512	#
9	F	74	54	20	10	77.5	Presbycusis	Right	Cochlear CI 512	#
10	M	80	60	20	18	87.5	Presbycusis	Left	Cochlear CI 512	dyslipidemia
11	F	6	0	6	5	113.75	Congenital	Right	MedEl Synchrony Pin	#
12	M	4	0	4	3	80	Congenital	Left	Cochlear CI 422	#
13	F	4	0	4	4	81.25	Congenital	Right	MedEl Concerto Pin	#
14	M	64	62	2	2	87.5	Presbycusis	Right	Neurelec	#
15	M	76	58	18	18	92.5	Presbycusis	Right	Advanced Bionics	#
16	M	54	0	54	45	112.5	Congenital	Right	Cochlear CI 512	#
17	M	85	45	40	40	115	Presbycusis	Right	Cochlear CI 512	#
18	M	45	7	38	38	91.25	Congenital	Right	Cochlear CI 512	#
19	M	31	29	2	0	107.25	Temporal bone fracture	Left	Cochlear CI 512	HIV
Mean (SD)		50.7 (28.4)	30.9 (27.6)	19.8 (19.2)	15.5 (16.1)	89.5 (24.2)				
Median		59.0	29.0	12.0	7.0	87.5				

CI: cochlear implanted; F = female; M = male; y = years; HA: Hearing aids; PTA = pure tone average threshold across 500, 1000, 2000, and 4000 Hz; HL: hearing loss; SD: standard deviation; dB: decibel.

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
