# Peer review of "Relationship between Metabolomics Profile of Perilymph in Cochlear-Implanted Patients and Duration of Hearing Loss"

_metabolites, 2019, doi:10.3390/metabo9110262_

Round 1

Reviewer 1 Report

Dear Sir/Madam,

I found your study fascinating.  However, beyond the demographic profile presented, I think it would be best for you to consider adding additional data.
For example:

1) sidedness of surgery

2) medications at time of surgery (steroids, antibiotics)

3)type of implant

4) comorbidities (diabetes, hypertension, smoking, cardiovascular disease, hyperlipidemia, etc.)

Reviewer 2 Report

In this study, the authors carried out metabolomic analysis of cochlear perilymph harvested from 19 subjects during cochlear implantation surgery. In total, they identified 106 metabolites by means of LC-HRMS. Moreover, they found significant difference in metabolomic profile between subject groups with ≤12 and >12 years of hearing loss. These results suggest that change of metabolic dynamics is involved in pathology of hearing loss.   

This manuscript for the first time reports the results of the metabolomic analysis of human perilymph. The observations would provide valuable insights for basic and clinical research fields. Nevertheless, the authors did not show any datasets of LC-HRMS. Furthermore, description of pathway analysis is insufficient. Although the work is of interest and importance, the manuscript is not acceptable in the current form.

Due to lack of datasets of LC-HRMS analysis including quantification of metabolites in each subject, I could evaluate neither validity nor quality of the results. I request the authors to show the measurement data and the instrumental parameters in the experiments.

The method and result of pathway analysis are unclear. The authors should explain which data they applied and how they performed the analysis. In addition, more detailed description and explanation of the output data they obtained are necessary.

Author Response

Reviewer #2

Comments and Suggestions for Authors

In this study, the authors carried out metabolomic analysis of cochlear perilymph harvested from 19 subjects during cochlear implantation surgery. In total, they identified 106 metabolites by means of LC-HRMS. Moreover, they found significant difference in metabolomic profile between subject groups with ≤12 and >12 years of hearing loss. These results suggest that change of metabolic dynamics is involved in pathology of hearing loss.   

This manuscript for the first time reports the results of the metabolomic analysis of human perilymph. The observations would provide valuable insights for basic and clinical research fields. Nevertheless, the authors did not show any datasets of LC-HRMS. Furthermore, description of pathway analysis is insufficient. Although the work is of interest and importance, the manuscript is not acceptable in the current form.

Due to lack of datasets of LC-HRMS analysis including quantification of metabolites in each subject, I could evaluate neither validity nor quality of the results. I request the authors to show the measurement data and the instrumental parameters in the experiments.

 Thank you for your revisions.

--> We apologize for the lack of datasets, we added in the methodology section the instrumental parameters of LC-HRMS described in Mavel et al. l. 235-259 “LC-HRMS analysis was performed using a UPLC Ultimate 3000 system (Dionex), coupled to a Q-Exactive mass spectrometer (Thermo Fisher Scientific, Bremen, Germany) and operated in the positive (ESI+) and negative (ESI−) electrospray ionization modes (one run for each mode). The perilymph analysis method, including the protocols for metabolite extraction and MS methods, are fully described in Mavel et al. [17]. Two liquid chromatography was performed using a Phenomenex Kinetex 1.7 μm XB – C18 column (100 mm x 2.10 mm) maintained at 40°C and a hydrophilic interaction liquid chromatography (HILIC) column (100 mm x 2.10 mm, 100 AÌŠ). The injection volume used for LC was 10µL, obtained after the preanalytical step including metabolome extraction, solvent evaporation and resuspension for LC analysis.

Metabolism of perilymph was quenched and metabolites were extracted with 100µL of methanol added to 0.8µL of perilymph. Samples were vortexed for 1min and stirred for 10min at 4°C. The samples were left at -20°C for 30min then centrifuged at 15000g for 15min at 4°C. The supernatant (95µL) was evaporated with a SpeedVac concentrator at 40°C and the dry residue was re-suspended in 100µL of methanol/water (50:50). Finally the samples were vortexed for 1min and centrifugated at 20000g at 4°C for 10min. Quality control (QC) samples were prepared from a mix of all extracted samples.

All metabolites described here were identified using a library of standard compounds (Mass Spectrometry Metabolite Libraryâ - IROA) by HPLC injection in the same analytical conditions. Identifications were done based on the retention time (±20 seconds of the standard reference), the molecular and fragment masses (within a range of 10 ppm) using high resolution, thanks to the orbitrap technology. The signal value was calculated using Xcalibur® software (Thermo Fisher 13  Scientific, San Jose, CA) by integrating the chromatographic peak area corresponding to the selected metabolite. Metabolites levels were normalized by the sum of all the metabolites of the profile, then data were log-transformed and autoscaled (mean-centered and divided by the standard deviation of each variable).”

The measurement data and datasets including quantification of metabolites are sent in the attachment tables named data_original.

The method and result of pathway analysis are unclear. The authors should explain which data they applied and how they performed the analysis. In addition, more detailed description and explanation of the output data they obtained are necessary.

--> We are sorry for the lack of details, we added l. 279-292 “Pathway impact represents a combination of the centrality and pathway enrichment results; higher impact values represent the relative importance of the pathway, relative to all pathways included in the analysis. Metabolic pathways are represented as a network of chemical compounds with metabolites as nodes and reactions as edges. Major criteria are used to perform an informative analysis regarding the quality of pathway data. These data were downloaded from the KEGG database [19] [20]. Chemical compounds and pathway topology information were obtained from the KEGG graph package [21] and the current library contains 874 metabolic pathways from humans. The KEGG pathway database (http://www.genome.jp/kegg/pathway.html) was used with the Metaboanalyst tool to explore the highlighted metabolic pathways. To focus on the most relevant data, we highlighted only the pathways that had significant Holm p-value <0.05 and a pathway impact >0.05. The pathway impact value was calculated as the sum of importance measures of the metabolites, normalized by the sum of importance measures of all metabolites in each pathway [20].”

However, we did not discuss the pathways because the impact values did not reach a significance. We added l. 122-124 “The main pathways involved in the discrimination between groups were methylhistidine, arginine and proline metabolism as well as taurine metabolism but their impact value did not reach significance neither.”

Reviewer 3 Report

Comment to metabolites-592967

The authors aimed to study the relationship between perilymph metabolite profiles and duration of hearing loss in a group of 19 patients receiving cochlear implants. Although this is a promising and challenging approach to study the pathophysiology of sensorineuronal hearing loss, the major problem of this study is the large variation in age of the patients (range: 4 – 86 y), 6 different underlying etiologies and the large difference in mean age between the patients groups with <12 years (N=10,  mean: 38 years of age) or >12 years hearing loss (N= 9, mean: 65 years of age). 

Using multivariate analysis, the authors showed that “age of implantation” was the second most important discriminant parameter to distinguish the two groups (<12 or >12 years hearing loss) (results section 2.2, line 96 and fig. 1)  Thus, considering the small group sizes (9 and 10 patients), the large difference in mean age and the significant contribution of age separating the two groups, one cannot conclude that differences in metabolite profiles are related to duration of hearing loss. Did the authors try to correlate duration of hearing loss (without dividing the patients in two groups) with abundance of selected metabolites? Did the authors consider statistical procedures to correct for the effect of aging? 

It has been suggested that metabolomics studies should include at least 20 patients per group. (Nyamundanda, et al. Bioinformatics 2013, 14, 338).

In the introduction there is no reference to the study of Mavel et al. (Ref. 17: Hearing Res 367 (2018), 129-136). The Mavel study is first mentioned in materials and methods section 4.3, although it seems that data of the same patient group are used in the present manuscript.

In section 3.1 of the Discussion it is stated that “this work is the first metabolomic analysis performed on human perilymph samples”, but reading Mavel et al. 2018, using the same patient recruitment protocol (2015-045), it seems to me that perilymph metabolite profiles were already reported here, without correlating the data with the duration of hearing loss.

In the introduction, describing the use of LC-HRMS in metabolomics (lines 71-73) a reference is made to Blasco et al 2014 (ref. 14). In this Neurology  paper 1H-NMR is used and not LC-MS.

Previous experimental studies on metabolite profiling in perilymph (guinea pigs, ref. 24, but also Fransson et al. Front Cell Neurosci 2013) should be mentioned in the introduction. 

Results section 2.1 and Table S1: 106 metabolites identified.

How were metabolites identified?

It is custom to validate that the metabolites reported are indeed the correct metabolites and not some other potential isomers. The authors should describe how the metabolites were annotated – and at which level according to the Metabolomics Standards Initiative (MSI) -, based on comparing retention times with synthetic standards, co-elution of fragments and/or accurate mass.  

Results section 2.3. ROC curve. It does not make sense to include “age at implantation” in the ROC curve, since such a curve should be used to assess the diagnostic potential of possible (novel) biomarkers.

Discussion section 3.1.

Lines 143-144: Which metabolites were changes in perilymph after loud noise

Line 148: First metabolomics study in human perilymph: Mavel et al. 2018 

Discussion section 3.2.

Lines 158-159: What does this mean? Please rewrite.

Discussion section 3.3.

Lines 172-73:  If N-acetylneuraminate  is a marker of cell death, I would expect elevated levels during ongoing cell death, i.e. in the beginning of progressive hearing loss and not after a long duration of hearing loss.

Lines 174-175: In which way were arginine and proline metabolism altered?

Material and methods section 4.1.

Table 1: Patients with presbycusis (n=5) are not mentioned in the table, but instead 5 patients with progressive HL, which is not mentioned in the text (lines 203-205). In Mavel et al. 2018 the same patient groups are described and also a group with progressive hearing loss with unknown etiology (n=4), which is not mentioned in this manuscript. Please clarify.

Material and methods section 4.2.

Line 224: around 1 µL of perilymph was collected. Be more precise. In Mavel et al. 0.8 µL to 2 mL was collected. Lumbar puncture refers to taking a CSF sample from the lower spine (spinal tab) and should not be used to sample perilymph from the inner ear.

Line 227:  are 1 µL samples aliquoted? How much was used for LC-HRMS?

Material and methods section 4.3.

Metabolites were extracted with 100 µL MeOH, but what was the perilymph sample volume?

Which types of HPLC columns were used ?

Round 2

Reviewer 2 Report

The authors adequately addressed my concerns. 

Author Response

Thank you for your comments and suggestions.

Reviewer 3 Report

The revised version of this manuscript is a significant improvement. The authors have included most of my suggestions, but the problems remains that population sizes are small and that there is a large variability in age of patients  and etiology of hearing loss. I understand, that the authors cannot include more cases in the current study, but I would like them to have a look at my remaining comments and suggestions.

Line 152: … to correlate a clinical criteria – correct this, for instance: to correlate with clinical data

Line 160: …altered in some phenotypes of patients defined by according to duration of hearing loss. – delete “defined by”

Line 117-118: 2.3. Legend: VIP: Variable Influence of Projection ; OPLS-DA : Orthogonal Partial Least Squares Discriminant Analysis ; p : p-value ; FC : fold change – The abbreviations should be written in the legend of Fig.1, but it is written as a separate heading of paragraph 2.3.

Discussion section 3.2.

Line 162: The discrimination between groups was correct. However, the distribution of patients into each group was also very different.  – I still do not understand what you mean, please rephrase.

Discussion section 3.3.

Lines 172-176:  If N-acetylneuraminate  is a marker of cell death, I would expect elevated levels during ongoing cell death, i.e. in the beginning of progressive hearing loss and not after a long duration of hearing loss.  Please discuss this point in more detail.

Lines 177-179: In which way were arginine and proline metabolism altered?

Table 1: Sidedness of CI change to: Site of CI

4.6. Pathway analysis

Line 277-278:  Based on both, to the size of population and the number of metabolites detected, we did not use the analysis of pathway impact to discuss our VIP [20].  – Why do you write a whole paragraph on analysis of pathway impact, when you did not use it to discuss the VIP scores shown in figure 1?

If there are good reasons not to use it, because of small population size and low numbers of metabolites detected, you can just state that analysis of pathway impact was not possible.

However, in Results section 3.3.  you state: “Arginine and proline metabolism were one of the most relevant pathways in this study” – how did you calculate this? Did you use the analysis of pathway impact or not?
